# A New Subgenus of the Genus *Phenolia* (Coleoptera, Nitidulidae) from Myanmar Cretaceous Amber with Taxonomic, Phylogenetic and Bionomic Notes on the ‘Nitidulid’ Group of Families [note 1]

**DOI:** 10.3390/insects14070647

**Published:** 2023-07-18

**Authors:** Alexander Georgievich Kirejtshuk, Josh Jenkins Shaw, Igor Sergreevich Smirnov

**Affiliations:** 1Zoological Institute, Russian Academy of Sciences, Universitetskaya Emb., 1, 199034 St. Petersburg, Russia; igor.smirnov@zin.ru; 2Natural History Museum of Denmark, Universitetsparken 15, 2100 Copenhagen, Denmark; josh.shaw@snm.ku.dk

**Keywords:** Apophisandridae, Boganiidae, Kateretidae, Parandrexidae, Smicripidae, new genus, new species, lectotype designation, anthophagy, pollinophagy

## Abstract

**Simple Summary:**

Studies of many fossil insects, in contrast to those from the Recent fauna, have considerable complications, as many structural features are missing or not clearly observable. Not infrequently, researchers are met with unusual insects, which are very different to those known from the present surrounding biota. Sometimes, in order to obtain reasoned arguments, they need to study many fossil specimens, in which unfamiliar structural features could be visible in different specimens with various degrees of preservation. A new supraspecific taxon for a new species from one of the very archaic nitidulid groups that putatively originated during the Cretaceous (Albian/Cenomanian Burmese amber, North Myanmar) is described in the present paper. For this reason, it was required to re-estimate some new data on fossil discoveries of the families, which can be regarded as closely related: Apophisandridae stat. nov., Kateretidae, Nitidulidae, Parandrexidae, Smicripidae and possibly Boganiidae. The new analysis makes it possible not only to describe *Phenolia* (*Palaeoronia*) *haoranae* subgen. et sp. nov., but also to propose one new generic taxon (*Antirhelus* gen. nov. for *Heterhelus buzina*) and two subfamily taxa (Antirhelinae subfam. nov. and Cretoparacucujinae subfam. nov.); in addition, new characteristics have been found in the external structures and genitalia of both sexes, which can be interpreted as more or less reliable evidence of the close relationship between the abovementioned families. The important characteristics of these families are found in their tendency to form and develop their interactions with generative organs of the higher plants, i.e. formation of feeding on them from the initial fungal diet. All these families reveal pollinophagous (anthophagous) lifestyles in part of their adult members, or they completely consist of pollinophagous (anthophagous) adults. Further, larvae of some of these members could be becoming anthophagous or carpophagous, and the last stage of these trophic changes seem to present the formation of a trophic imaginal and larval regime with the use of plant vegetative organs.

**Abstract:**

A new subgenus, *Palaeoronia* subgen. nov., is described from the Cretaceous amber of North Myanmar (Kachin State) and assigned to the genus *Phenolia*. The type species of the new subgenus, *Phenolia* (*Palaeoronia*) *haoranae* subgen. et sp. nov., is characterized by a rather ‘archaic’ aspect. A discussion of the diagnostic and structure of the *Soronia*-complex of genera (together with the *Phenolia*-complex of genera) (Nitidulinae, Nitidulini) is proposed. Reasons for the ‘conservatism’ of this group during the Mesozoic and Cenozoic are discussed. The position of the Apophisandridae stat. nov. (type genus *Apophisandra*) and the transfers of the following genera into this family: *Cretaretes, Electrumeretes, Furcalabratum, Pelretes*, Polliniretes, Protokateretes, *Protonitidula*, and *Scaporetes,* from the Kateretidae, Nitidulidae or Cerambycidae are grounded. The relations of the family Parandrexidae (with inclusion of the genus *Cretoparacucujus,* transferred from Boganiidae with a proposal of the subfamily Cretoparacucujinae subfam.nov.), *Martynoposis* and *Parandrexis* are considered. The genus *Antirhelus* gen. nov. (type species *Heterhelus buzina*) is assigned to the new subfamily, Antirhelinae subfam. nov. in the family Kateretidae. The fossil records of the ‘nitidulid’ group of families (Apophisandridae stat. nov., Kateretidae, Nitidulidae, Parandrexidae, Smicripidae and possibly Boganiidae) are reviewed. The relationship of the family Boganiidae, some aspects of pollination and pollinophagy, and also changes in beetle diet in the past are discussed. The lectotype of *Parandrixis parvula* is designated.

## 1. Introduction

The family Nitidulidae is represented in the Recent fauna by nine subfamilies and a moderate number of species ([1,2], etc.), which are rather diverse in terms of structure and bionomy, and they also share some similarities with many other cucujoid families. In recent decades at the end of the 20th and beginning of the 21st centuries, most researchers regarded the three families, Nitidulidae, Smicripidae and Kateretidae, as closely related because of many similarities in both imaginal and larval structures. Therefore, the idea to consider them as a separate superfamily Nitiduloidea appeared [3,4,5]. Also, the nitidulid subfamily Cybocephalinae was sometimes treated as a separate family ([6,7], etc.). Many hypotheses were proposed for a relationship between the ‘nitidulid’ group of families with other cucujoids ([5,8,9], etc.). A large group of co-authors [10] recently summarized some models of relationships within cucujoid families and proposed a compromised scheme of relationships without a clear definition of their concept, joined many families in a new version of the superfamily Nitiduloidea and provided it with an obscure diagnosis. This version would be better treated as very premature and requires a more detailed and comprehensive analysis of many cucujoid groups. The current preliminary composition of Nitiduloidea *sensu stricto* proposed as the ‘nitidulid’ group of families seems to have more certain traces of a joint phyletic root than that proposed by the latter group of co-authors (see the below discussion).

The fossil record of Nitidulidae is not well documented, although many specimens are accessible for study and description in many scientific collections. Until now, fossils of the subfamilies Epuraeinae, Cillaeinae, Cryptarchinae, Cybocephalinae, Meligethinae, and Nitidulinae ([11,12,13,14,15,16,17,18,19,20], etc.) have been described. The tribe Nitidulini is at least represented in the current fossil record by the genera *Menatoraea* Kirejtshuk and Nel, 2018 [18], *Microsoronia* Kirejtshuk and Kurochkin, 2010 [14], *Omositoidea* Schaufuss, 1892 [13,21], *Palaeolycra* Kirejtshuk and Nel, 2018 [18], *Palaeometopia* Kirejtshuk and Poinar, 2007 [13], *Phenolia* Erichson, 1843 [11,22] (“*Lasiodactylus*”), *Prometopia* Erichson, 1843 [13,22], *Soronia* Erichson, 1843 [18,22], *Sorodites* Kirejtshuk, 2018 [17], while the mentions of other genera are dubious or erroneous ([11,18], etc.). The genus *Phenolia* is known in fossils by only three species from the Eocene Baltic amber [14], Eocene/Oligocene of Bembridge Marls (England) [19] and Middle Miocene of Karagan (Stavropol Region, Russia) [11].

A new nitidulid species, *Phenolia* (*Palaeoronia*) *haoranae* subgen. et sp. nov. (Figure 1 and Figure 2), is discovered and described from an amber piece from the Cretaceous Burmese amber. Among other syninclusions of the fossil under current description of a male probably belonging to the genus *Mesosmicrips* Kirejtshuk, 2017 [23] (Figure 3A), a taxon based on one female also from the Cretaceous Burmese amber [24] is found. The abovementioned genus belongs to the family Smicripidae in the ‘nitidulid’ group of families. Other fossils of this family will be considered below. In order to reach a clear understanding of the situation in the recorded fossils of the ‘nitidulid’ group of families, it is necessary to review the currently available data on fossils of Kateretidae and Smicripidae, re-arrange their members, and also re-estimate the position of the Parandrexidae, which is sometimes regarded as uncertain, although at present almost no doubt that it is closely related to the ‘nitidulid’ group of families. Therefore, in the discussion below, the position of the Parandrexidae is re-grounded, and a new subfamily, Cretoparacucujidae subfam. nov., is proposed within this family for the genus *Cretoparacucujus* Cai, Escalona, Li, Huang, and Engel 2018 [25], initially placed in Boganiidae. Some fossil ‘kateretid’ genera are currently described [26,27,28,29,30,31,32], and some of them are tentatively considered as pollinators. At the time, some probable evidence of plant counter adaptation for ‘cantharophily’ in the description of a cockroach nectarivory was published [33]. These ‘kateretids’ should indeed be assigned to a separate family Apophisandridae stat. nov., which was originally proposed as the tribe Apophisandrini Molino-Olmedo, 2017 [34] within the family Cerambycidae. One fossil genus recently described as a nitidulid, *Protonitidula* Zhao, Huang and Cai, 2022 [35], also belongs to the Apophisandridae stat. nov. Apophisandridae stat. nov. and Parandrexidae (Figure 3B–D and Figure 4) are re-described and compared with the related taxa. Another fossil from the Eocene Rovno amber, described as a member of genus *Heterhelus* Jacquelin du Val, 1858 [36], needs to be transferred to the new genus *Antirhelus* gen. nov. within the family Kateretidae. The new subfamily Antirhelinae subfam. nov. is here proposed to accommodate it.

Studies of fossils including the present description of *Palaeoronia* subgen. nov., and other new data on different cucujoids make it possible to re-estimate the position of some families with the application of some key principles used to elaborate a system of the family Nitidulidae, including grounding the isolation of Kateretidae, which was previously regarded as a subfamily among nitidulids [37]. First, after the widest possible comparison of the genital structures of both sexes, a division and grouping of supraspecific taxa was made, which did not correspond to the groupings in the previous classifications, made mainly according to some external characteristics [37].

Furthermore, many nitidulid groups were living during the same periods with partial or complete anthophagy and carpophagy, the placement of which in the previous systems does not correspond with either simple sorting according to external structural characteristics or sorting according to their genital structure. Another key principle of the studies conducted by the senior author of this paper, a principle used at the beginning of each study, was an analysis and comparison of larval inhabitance and diet, which can either fit or differ from these structural peculiarities of adults. As a result, a scheme of regular changes in inhabitance and feeding of both active ontogenetic stages, larvae and adults (see the Discussion section below and Figure 5 borrowed from [38]), which can indeed be applicable for Nitidulidae and closest families, as many cucujoid (or cucujiformian) groups have an expressed tendency to include regular phytophagy, particularly larval phytophagy. These studies, through the combination of these two principles, resulted in the system of Nitidulidae [1]. This was a reason to extrapolate them on fossils, although, of course, the estimation of bionomical peculiarities of the latter is a rather delicate matter and, in many cases, more or less hypothetical.

## 2. Material and Methods

The specimen (holotype of *Phenolia* (*Palaeoronia*) *haoranae* subgen. et sp. nov., ((BJFC)HE20230001) for study was provided by Haoran Sun, a PhD student at the Natural History Museum of Denmark and Beijing Forestry University. It will be deposited to the collection of the Beijing Forestry University. This amber piece contains coleopteran and mecopteran inclusions and originates from the amber mines situated in the Hukawng Valley of Kachin State, Myanmar (26°21′33.41′′ N–96°43′11.88′′ E) (see Geological setting). The holotype of the new species was studied using a Leica M165 C microscope (Leica Microsystems GmbH, Wetzlar and Mannheim, Germany). Photographs of this specimen were taken using two set-ups: a Canon EOS 6D DSLR (Canon Inc., Tokyo, Japan) with a Canon Macro Photo lens MP-E 65 mm or the same camera mounted on a Zeiss Axioskop 50 (Zeiss Group, Jena, Germany) via an LM Digital SLR Universal Adapter. Separate slides with the beetle images were stacked using ZereneStacker 1.04 (Zerene Systems LLC, Richland, WA, USA) and lightly edited using Adobe Photoshop. Drawings were prepared using Adobe Illustrator. For comparisons with other members of the families under consideration, the authors used fossil and modern species from the Zoological Institute of the Russian Academy of Sciences (St. Petersburg) and from Muséum national d’Histoire naturelle (Paris, France) and from other scientific collection over the world. These specimens were examined using a Leica MZ 12.0 stereomicroscope (Leica Microsystems GmbH, Wetzlar and Mannheim, Germany) with a DFC290 digital camera (Leica Microsystems, Wetzlar and Mannheim, Germany) at the Zoological Institute and an Olympus SCX9 stereomicroscope (Olympus Corporation, Tokyo, Japan) equipped with an Olympus camera at the Muséum national. The fluorescent microscopic images were taken at the Zoological Institute using a DM 6000 B microscope (Leica Microsystems GmbH, Wetzlar, Germany) with 2.5× and 5× objectives, Leica DFC 345 FX camera and Leica Application Suite 3.7 software with an Image MultiFocus module (Leica Microsystems GmbH, Wetzlar and Mannheim, Germany); the filter set applied was, in most cases, N21 or sometimes L5 (Leica Microsystems GmbH, Wetzlar and Mannheim, Germany).

The data considered in this paper are obtained mostly from paleontological materials and studied by traditional comparative morphological methods. Further studies of the groups here considered in different aspects of their phylogeny will probably allow us to reach a comparable level of knowledge in each aspect and fulfill a complete analysis of all data in accordance with the principle of multiple parallelism [39]. Therefore, this paper mostly applies to the characteristics visible in the fossil specimens. In addition, parallel changes seem to be one of general evolutionary regularities in the groups of organisms, and their manifestation can be traced in a greater expression in more closely related organisms. This is expressed in the order Coleoptera in general [4] and also the related families considered in this paper.

### 2.1. Depositories

BFU—Beijing Forestry University, Beijing, ChinaCNU—Capital Normal University, Beijing, ChinaGPIH—Center of Natural History (formerly Geological–Paleontological Institute and Museum), Hamburg, GermanyVSEGEI—Russian Geological Research Institute (VSEGEI), St. Petersburg, RussiaZIN—Zoological Institute of the Russian Academy of Sciences, St. Petersburg, Russia

### 2.2. Geological Setting of the Locality of the New Species

The amber piece ((BJFC)HE20230001 (BFU)) with inclusions originated from mines in the Hukawng Valley in the state of Kachin in Myanmar. The fossil resin has been dated stratigraphically and radiometrically from late Albian to early Cenomanian [40]. According to the traditional interpretation [41], the probable Cenomanian radiometric age of Burmese amber has been estimated as being taken from sedimentary beds, indicating that it had been re-deposited. Much of the resin is rolled and bored by pholadid bivalves, which demonstrates that the resin was hard before it was buried [42]; the centers of the resin pieces, however, were still soft when bored, so the formation of the amber is considered to be contemporaneous with the deposition of the bed [43,44]. Thus, the age of this amber still remains unclear, although an enormous concentration of amber with inclusions in certain geological layers gives us evidence that its deposition occurred under peculiar conditions and during a definite term. Mao et al. [45] alleged a primary deposition of this amber in marine circumstances; however, it is still impossible to explain a very considerable accumulation of resin under such conditions without the essential addition of fossil resin transported by river water from large territories. Considering these circumstances, the amber material from this locality has to be considered approximately coeval with the amber-bearing deposits, which is, at its earliest, Cenomanian in age. Nuclear magnetic resonance spectra and the presence of araucaroid wood fibers in amber samples from the Noije Bum 2001 Summit site indicate an araucarian (possibly *Agathis*) tree source for the amber (e.g., [46,47], etc.).

## 3. Results

### Systematic Palaeontology

Order Coleoptera Linnaeus, 1758Suborder Polyphaga Emery, 1886 [48]Infraorder Cucujiformia Crowson, 1960 [49]Family Nitidulidae Latreille, 1802 [50]Subfamily Nitidulinae Latreille, 1802 [50]Tribe Nitidulini Latreille, 1802 [50] **Genus *Phenolia*** Erichson, 1843 [22]

**Remarks**. This genus was revised for the Afro-Madagascan modern congeners with a review of distinguishing characteristics of its subgenera and similar genera in the Recent world fauna [51] from the *Soronia* complex of genera including taxa from the former *Phenolia* complex [1]. Lawrence [52] criticized this subgeneric structure of the genus and proposed to raise the subgenera of *Phenolia* to separate genera; however, the author used for his analysis mostly the characteristics which are not easy to observe, and he studied the Australian species and a few modern species from other areas without taking into consideration the scope of variation in the characteristics among congeners from other zoogeographical regions. As this mentioned publication includes some pictures of small structures, it can be used to illustrate variability of structures rather than diagnostic characteristics for the taxonomic discrimination of the groups considered in it. Nevertheless, the structure of the prosternal process, the outline of postmetacoxal lines of the abdominal ventrite 1, possibly the stridulatory files on prohypomera and other structural features can be used for group splitting in the *Soronia* complex of genera in the new sense (see the Discussion section below). In general, this paper is useful for a wider comparison of every characteristic among many representatives from different regions and particularly species of *Lasiodites* Jelínek, 1999 [53]. On the other hand, characteristics of the new subgenus and unpublished data make it possible to join the *Soronia* and *Phenolia* complexes of genera in the previous sense, because separate species of the genera *Lobiopa* Erichson, 1843 [22], *Microsoronia* [14], *Soronia* and subgenus *Lasiodites* demonstrate some characteristics which are extremely similar, while in other members of the mentioned groups, they are quite different (see the Comparison of the new subgenus and Discussion section below).



**Subgenus *Palaeoronia*** Kirejtshuk and Jenkins Shaw, **subgen. nov**.


http://zoobank.org/urn:lsid:zoobank.org:act:184394EB-684D-4D9F-ACA0-74E2813E334B


Type species: *Phenolia* (*Palaeoronia*) *haoranae* sp. nov.; fossil, Burmese amber, Cretaceous, Albian/Cenomanian.

**Etymology.** The name of this new subgenus is formed from the Greek ‘πᾰλαιός’ = ‘palaeo’ (ancient, older) plus the generic root ‘*ronia*’ (*Hisparonia*, *Pleoronia, Sinosoronia*, *Soronia*, etc.). Gender feminine.

**Diagnosis**. Body oval (circa 2.5 mm long); vestiture uniform, sparse and rather short, without both cilia and longer hairs forming longitudinal rows; punctures on elytra very coarse, rather shallow; frons with lateral dilatations arcuately curved posteriorly and joined with inner edge of eye at distance subequal to one-third of eye longitudinal diameter; antennal grooves arcuately convergent posteriorly; antennal club subovoid and widest at base; pronotum widely explanate at arcuate sides; elytral surface even, without trace of regular longitudinal sculpture, broadly arcuate at widely explanate sides and conjointly arcuate at apices; male anal sclerite well exposed from under apex of pygidium; prosternal process arrow-shaped and comparably wide with antennal club, slightly curved along procoxae; mesoventrite about as long as metaventrite; submetacoxal lines following posterior edge of metacoxal posterior edge.

**Comparison**. The species under description is similar to all subgenera of *Phenolia*, although the most similarity seems to be found between *Phenolia* (*Palaeoronia*) *haoranae* subgen. et sp. nov. and some separate members of *Lasiodites*. Nevertheless, the combination of the general shape of a small and wide body of the new species, very sparse and very shallow elytral punctures, very fine dorsal pubescence without a trace of longitudinal rows on elytra, and even integument of the elytra is rather distinct from all known congeners. Also, it is important that, in contrast to other *Phenolia* subgenera, the mesoventrite of the type species *Palaeoronia* subgen. nov. is very long, and its scutellum is subtrapezoid, with subtruncate apex, and also the elytral integument with very coarse punctation. Nevertheless, the elytral sculpture of species of other subgenera can be rather obliterated and with unclear outlines of very shallow punctures and, therefore, slightly reminiscent of that of the new subgenus. Some species of other *Phenolia* subgenera (as exceptions to some extent) have almost not traced both costae and striae on elytra. Another important aspect is that most characteristics mentioned to discriminate the new subgenus are present in separate *Lasiodites* species, but members of the latter subgenus never have so small and so wide a body.

Other characteristics distinguishing the new subgenus from other subgenera of the genus are as follows:-From *Aethinodes* Blackburn, 1891 [54]: in the arcuately convergent antennal grooves and postmetacoxal lines following the posterior edge of the metacoxal posterior edge;-From *Phenolia* [22]: in the moderately wide prosternal process with arrow-shaped apex (not greatly widened apex of the prosternal process before the slightly arcuate apex);-From *Plesiothina* Kirejtshuk, 1990 [55]: in the not-modified antennal club (elongate oval and widest at base) and elytral apices (not-widened apically antennal club and conjointly projecting elytral apices).

On the other hand, *Phenolia* (*Palaeoronia*) *haoranae* subgen. et sp. nov. demonstrates some features characteristic of members of *Microsoronia* from the former *Soronia* complex, also having a comparatively small body with widely explanate sides of pronotum and elytra, diffuse and frequently coarse elytral punctation, and so on. Nevertheless, the new species is quite distinct from almost all members of *Microsoronia*, with the pronotal sides gently narrowing towards posterior angles without basal emargination or excision, the absence of a trace of longitudinal rows of longer hairs on elytra (although *Microsoronia interfax* Kirejtshuk and Kurochkin, 2010 [14] also has pronotal sides gently narrowing to posterior angles), a comparatively narrow prosternal process with almost arrow-like apex, a markedly shorter metaventrite than in the congeners, and wide tarsi (particularly anterior ones).

**Composition**. The type species only.



***Phenolia* (*Palaeoronia*) *haoranae*** Kirejtshuk and Jenkins Shaw, **sp. nov.**


http://zoobank.org/urn:lsid:zoobank.org:act:CA954FCF-360D-47E2-957A-B5B6F2FD81E5


**Type material.** Holotype ((BJFC)HE20230001, BFU), male, a complete and well-preserved beetle, is included in a flat oval amber piece (20 mm × 13 mm) (Figure 1A,B). The plane of the beetle body is somewhat declined regarding both upper and lower planes of the amber piece. This amber piece with the holotype also included one male beetle probably from the genus *Mesosmicrips* (Smicripidae) and two males of *Parapolycentropus* sp. (Mecoptera, Pseudopolycentropodidae), one nematoceran specimen (Diptera), many remains of some very small beetles apparently with a body around 1.0 mm (two elytra: one of oval beetle and another of elongate beetle, some separate legs of very small beetles, two fragments of abdomens, part of head with clubbed antennae, etc.), one piece of very dark (almost blackish) organic matter comparable in size with the holotype of the nitidulid under description, and some small oval pieces of reddish organic matter spread in different places of the amber piece.
Figure 1*Phenolia* (*Palaeoronia*) *haoranae* subgen. et sp. nov. (holotype, (BJFC)HE20230001) (BFU), male; Albian/Cenomanian Burmese amber: (**A**)—amber piece with the body, ventrally; (**B**)—idem with the body, dorsally; (**C**)—body, dorsally; (**D**)—idem, ventrally. Scale bars = 1.0 mm.
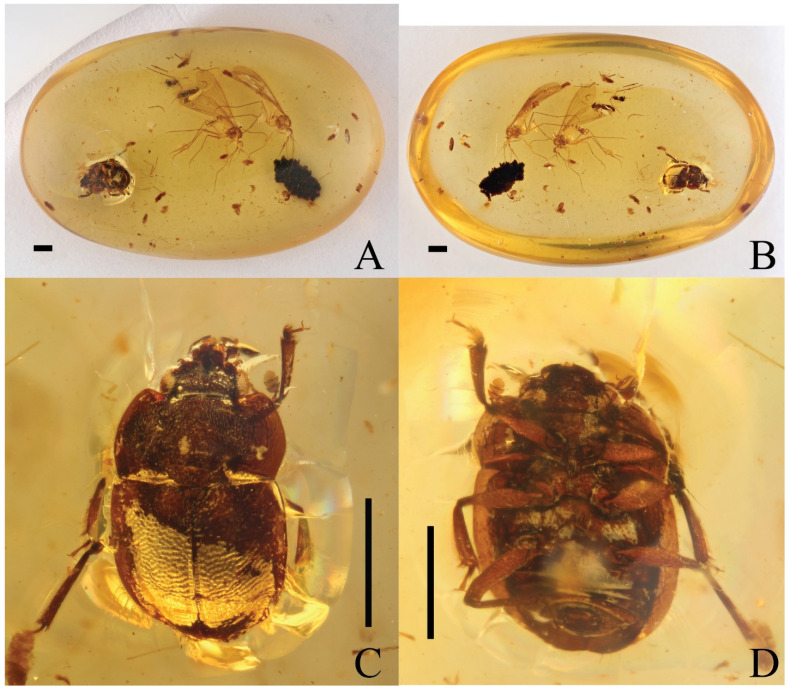


**Etymology**. The epithet of this new species is formed from the name of Haoran Sun, who generously provided the co-authors of this paper with the holotype of this species for their study.

**Description (male)**. Length of body more than 2.5, width 1.3 mm. Body oval (Figure 1C,D), less than twice as long as wide, moderately convex dorsally and slightly convex ventrally; subunicolorous reddish and with somewhat lighter sides of pronotum and elytra, and also appendages; covered with very fine, uniform, comparatively short and slightly conspicuous hairs, without trace of longitudinal rows (their length on dorsal sclerites somewhat greater than distance between their insertions).

Integument of head and pronotum (Figure 1C and Figure 2A) with more or less distinct punctures, about as coarse as eye facets, with interspaces between them somewhat less than one puncture diameter and well-raised microreticulation or partly obliterated. Integument of elytra with coarser, shallow and less distinct punctures, apparently about twice as coarse as those on head and pronotum, with shallow surface and interspaces between them rather smooth to alutaceous. Integument of underside of head, thorax and abdominal ventrite 1 with punctures about as coarse as those on the upper surface of head and pronotum, but sparser and with more or less obliterated surface.
Figure 2*Phenolia* (*Palaeoronia*) *haoranae* subgen. et sp. nov. (holotype, (BJFC)HE20230001) (BFU), male; Albian/Cenomanian Burmese amber: (**A**)—head and prothorax with right leg, dorsally; (**B**)—antennal club, dorsally; (**C**)—apex of prosternal process and median part of pterothorax, ventrally; (**D**)—head, dorsally; (**E**)—abdominal apex, posteriorly; (**F**)—mentum, labial palpi, right maxillary palpus, genal processes and inner edges of antennal grooves, ventrally. Abbreviation: scl—male anal sclerite. Scale bars: for A, C and D = 0.5 mm; for B, E and F = 0.2 mm.
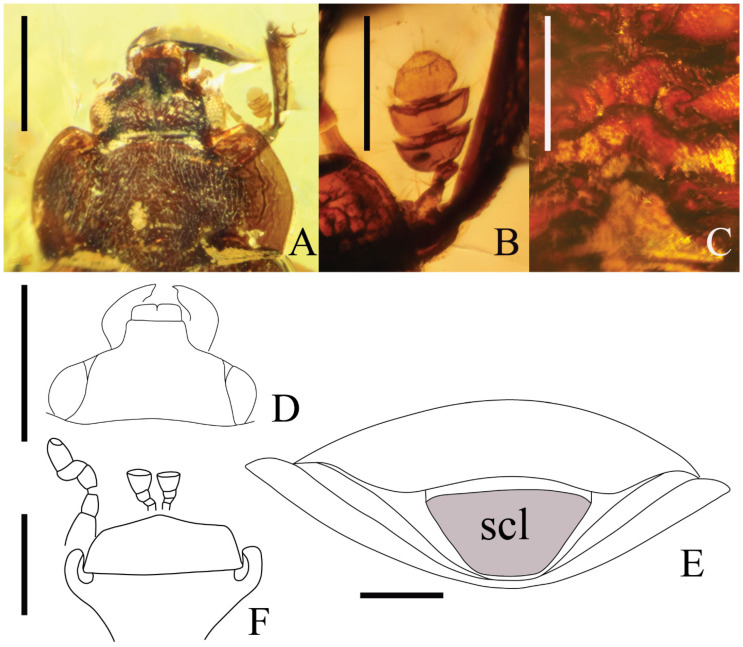


Head markedly shorter than distance between eyes and with weak transverse depression at lateral frontal dilatations over antennal insertions; lateral end of these dilatations arcuately curved posteriorly and joined with inner edge of eye at distance subequal to one-third of eye longitudinal diameter; eyes moderately large and moderately coarsely faceted. Labrum well exposed, subtransverse anteriorly and with expressed closed median excision. Mandibles moderately exposed, with comparatively sharp apices and with slightly raised subapical tooth. Labial ultimate palpomere cup-shaped, about as thick as long, widest at truncate apex. Maxillary ultimate palpomere somewhat elongate, about twice as long as thick, narrowing to obliquely truncate apex. Mentum subpentagonal, about 2.5× as wide as long (Figure 2F). Antennal grooves convergent posteriorly, with distance between their proximal ends about half as great as width of mentum. Antenna with scape of usual shape, moderately short pedicel, antennomeres 3–6 nearly twice as long as thick, and three-segmented club about one and half times as long as wide and rounded at apex. Pronotum moderately convex at disc and widely explanate at arcuate sides, anterior edge moderately bi-emarginate and with widely rounded anterior angles, and posterior edge very slightly bi-emarginate with distinct posterior angles. Scutellum subtrapezoid and with subtruncate apex. Elytra about 1.6 × as long as wide together, without trace of regular longitudinal sculpture, broadly arcuate at widely explanate sides and conjointly arcuate at apices. Pygidium widely subtruncate at apex, and widely subtruncate apex of anal sclerite well exposed from under apex of pygidium (Figure 2D, scl).

Prosternum moderately long, its process being arrow shaped and markedly wider than antennal club, slightly curved along procoxae. Mesoventrite slightly medially depressed (Figure 2C). Metaventrite rather short (Figure 2C), about as long as mesoventrite, distinctly widely depressed along middle and apparently with expressed discrimen, posterior edge between metacoxae moderately emarginate; posterior edge between metacoxae shallowly angularly excised. Distance between mesocoxae and that between metacoxae more than twice as great as that between procoxae (apparently nearly 2.5×). Abdominal ventrite 1 apparently longest; hypopygidium apparently widely emarginate at apex.

Legs of usual shapes and comparatively long (Figure 1C,D). Tibiae subtriangular and gradually widening apically; anterior one slightly wider than antennal club; intermediate one about as wide as antennal club, and posterior one somewhat narrower. Anterior tarsi nearly half as wide as corresponding tibia; intermediate ones about one-third as wide as corresponding tibiae, and posterior ones much narrower than latter; tarsal claws simple.

**Notes on probable bionomy**. A significant structural similarity of *Phenolia* (*Palaeoronia*) *haoranae* subgen. et sp. nov. to the members of *Lasiodites* makes it possible to admit for the new species a somewhat similar lifestyle and trophic regime. The modern members of the latter are mostly associated with decaying organic matter of plant origin, including various crevices with fungal availability (frequently under bark), soft fruits (sometimes dried ones) and other decaying plant remnants. They can inhabit many biotopes with trees and bushes spread in nearly all areas with a tropical and subtropical climate.

## 4. Discussion

The nitidulids from the Cretaceous amber, including ones awaiting description, represent some paleoendemic genera. The great similarity of the specimen described here to the modern members of the *Soronia* complex, which is united with the *Phenolia* complex, could be interpreted as evidence for stability of their structural type in the family Nitidulidae. It could be associated with some kind of conservatism in the habitats and bionomy of members of this joined complex. The elytral sculpture and other characteristics of the new species are similar to those in some modern species of *Soronia* and *Lobiopa* and particularly to *Microsoronia*. These similarities and other abovementioned ones give an additional argument to revise at least all the basal genera included by Kirejtshuk in 2008 [1] in the *Soronia* and *Phenolia* complexes of genera. On the other hand, many structural features of species of the *Soronia* complex in the new sense could be more or less generalized and traceable over long periods of time, fluctuating around a certain possible structural norm. In addition, the subfamily Nitidulinae could retain many archaic structural features in general, and it is likely that most structural variability of this subfamily is partly connected with this circumstance. Thus, older fossil representatives of this complex of genera can be expected to be discovered in the future. Another very archaic nitidulid group is Epuraeini, recorded from the early Cretaceous of Baissa, Transbaikalia (Buryatia), represented in the current fossil record by three species of *Crepuraea* Kirejtshuk, 1990 [11]. Both the *Soronia* complex of genera and *Crepuraea* could be close to basal nitidulid diversification of carpophiline and nitiduline lineages of the family.

These two phyletic lineages can be defined by two types of the male genitalia [37,38]. Males of the groups of the carpophiline lineage have ’trilobate‘ aedeagus, including the tegmen being deeply dissected medially (functioning as long lobes analogous (or homologous) to paramera) and having space between them for movement of the penis trunk, which is oval in cross section and not very compressed for easy movement between lateral lobes of the tegmen (when the penis trunk is moving posteriorly) ([37]: Figures 1г, д, е (in Russian) and 2a, б (in Russian). The nitiduline lineage have a ’bilobate‘ aedeagus consisting of a wide tegmen and penis trunk with both being dorsoventrally compressed, acting together when moving posteriorly and ‘opening’ for the inner sac of the penis trunk ([37]: Figures 1ж, з, и (in Russian) and 2в, г (in Russian). It does not matter what the origin of this divergence (genetic or morphogenetic) was, but at present, it is informative and can be regarded as plesiomophy (nitiduline lineage) and apomorphy (carpophiline lineage).

It is necessary to noted that the Nitidulidae, in contrast to both Kateretidae and Smicripidae, have a symmetric aedeagus with the fused paramera and phallobase, while Kateretidae and Smicripidae have an asymmetric aedeagus, with articulated paramera in Kateretidae and fused in Smicripidae ([5]: Figures 20–22 and [37]: Figure 1a, б, в (in Russian). Also, the nitidulid ovipositor gonocoxites usually bear more or less distinct or traced inner and outer lobes, while the kateretid and smicripid gonocoxites of the ovipositor are weakly sclerotized (nearly membranous), without clear separation on outer and inner lobes, and styli are located apically. Thus, these differences are rather important evidence of the divergence which seemed to happen very long before the divergence producing the nitiduline and carpophiline lineages. Nevertheless, there is considerable similarity in the habitus and many structures of these families, in most cases probably having a homoplastic origin, which apparently can be interpreted as evidence of their having a common phyletic root.

Another important aspect is that both Kateretidae and apparently fossil Apophisandridae stat. nov. have articulated paramera ([26]: Figure 4A) and a membranous ovipositor not divided into inner and outer lobes ([35]: Figure 1). The taxon Apophisandrini was proposed as a tribe in the family Cerambycidae (Chrysomeloidea) after the study of one badly preserved specimen (holotype of *Apophisandra ammytae* Molino-Olmedo, 2017 [34]), and later, it was transferred to the family ‘(?)’ Parandrexidae [56]. Poinar and Brown [26] described the next species of this group (*Furcalabratum burmanicum* [26]). Before the description of the latter, the researchers asked the senior author of the present paper about the possibility of its kateretid attribution. He agreed only that the species under description of the latter authors is rather similar to Kateretidae, although it is very peculiar and very different from the members of this family known at that time, which explains why it is difficult to assign it to the mentioned family. Firstly, this specimen has tarsi very distinct from those in all other taxa of the ‘nitidulid’ group of families from the recent fauna in the tarsomere 4, which is comparably sized and similar in shape with preceding ones (Figure 3B,D (and also [29]: Figure 1e,f), while the tarsomere 4 of all modern members of the ‘nitidulid’ group of families is markedly smaller and cylindrical or completely reduced (as in Cybocephalinae). There are many other characteristics of the apophisandrids mentioned in the Taxonomic Notes section below, which are also very different from those in all modern members of the ‘nitidulid’ group of families recognized at that time as being closely related. Nevertheless, Poinar and Brown [26] obtained support for the kateretid attribution from A.R. Cline, which they used to place the specimen under their description in Kateretidae. As a result, the species was published with this attribution, which should be regarded as erroneous in the current interpretation of the ‘nitidulid’ group of families (or Nitiduloidea *sensu stricto*). As this group (here treated a separate family Apophisandridae stat. nov.) is quite abundant among beetles included in Burmese amber, many researchers followed Poinar and Brown [26], and some publications appeared in a rather short period [27,28,29,30,31,32]. However, one genus of this group was recently described among Nitidulidae [35]. These circumstances need to be clarified in more detais (see below) in order to achieve a correct interpretation of the ‘nitidulid’ group of families in general and each family and subfamily in it. In addition to the already-described genera and species of Apophisandridae stat. nov., the senior author of this paper gathered dozens of specimens, which are scheduled to be studied and described in a comprehensive revision of this family.
Figure 3Families Smicripidae (**A**) and Apophisandridae stat. nov. (**B**–**D**): (**A**)—body of probable *Mesosmicrips* sp., male, syninclusion of the holotype of *Phenolia* (*Palaeoronia*) *haoranae* subgen. et sp. nov. (BFU), lateroventrally; Albian/Cenomanian Burmese amber; (**B**)—body of probable *Furcalabratum* sp. (GPIH, Carsten Gröhn collection: 11023), female; ventrally; Albian/Cenomanian Burmese amber; (**C**)—head of the same specimen, anterodorsally; (**D**)—body of specimen of unnamed genus and species, lateroventrally (ZIN, 0003). Scales bars for A = 1.00 mm, for B = 0.41 mm, for C = 0.22 mm, and for D = 0.44 mm.
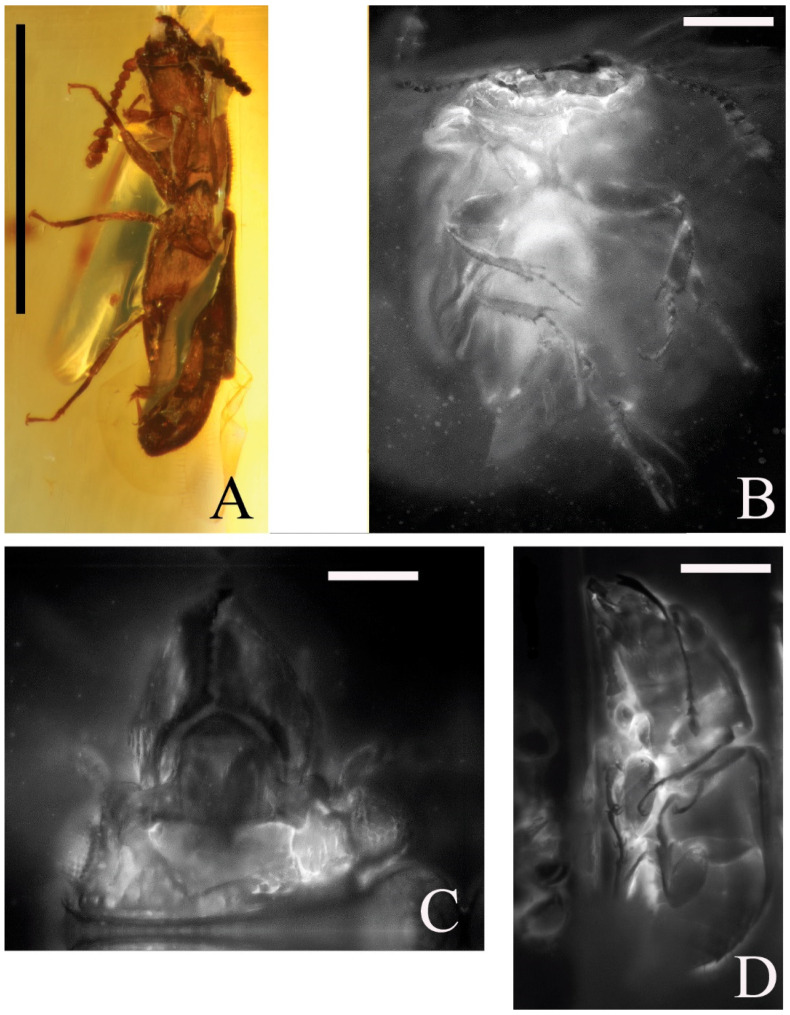

Figure 4Family Parandrexidae. (**A**)—body of the lectotype of *Parandrexis parvula* (Parandrexinae *sensu stricto*) (VSEGEI: 5017-1), ventrally, Karatau, late Jurassic, Karabastau Formation; (**B**)—body of specimen of unnamed genus and species (Cretoparacucujinae subfam. nov.) (CNU: Col-LB 2009825), dorsally, length of body circa 4.0 mm, early Cretaceous, Liaoning, Yixian Formation. Scale bar for A = 1 mm.
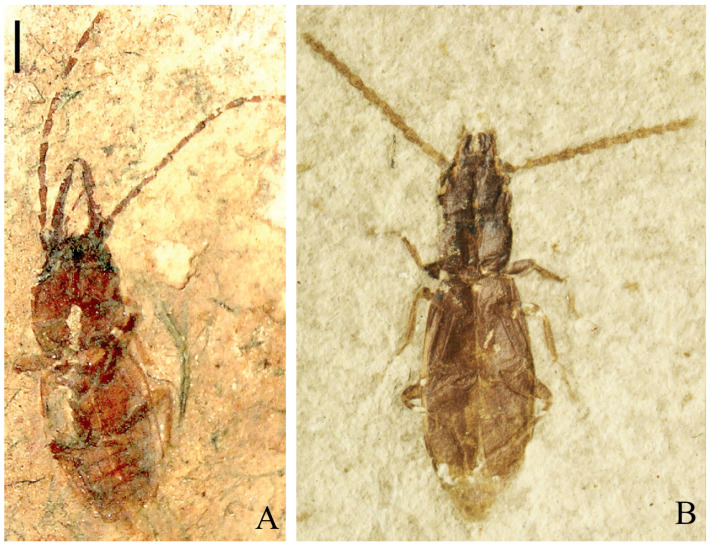



Regarding the genus *Antirhelus* gen. nov., which possesses the distinct diagnostic family syndrome of Kateretidae ([57]: Figures 1 and 2), it shows a very peculiar structural feature, which can be treated as a probable synapomorphy with two kateretid genera (*Kateretes* and *Antirhelus* gen. nov.) and which consists of the strongly modified two first antennomeres, expressed mostly in males of the first and in the holotype (female) of *Antirhelus buzina* (Kupryjanowicz, Lyubarsky and Perkovsky, 2021), comb. nov. [57]. However, in addition to this characteristic in antennal structure, the frons of the latter is so different from that of most modern members of the ‘nitidulid’ group of families (at least all known its modern anthophagous representatives) that it certainly needs to be considered as a separate suprageneric taxon (*Antirhelinae* subfam. nov.).

Crowson published some mentions of ‘*Parandrexis’*. In 1981 ([8]: 666), he wrote: ‘It is even possible that some of the modern beetles associated with cones of Cycadaceae are direct descendants of groups which once frequented Cycadeoid flowers, notable examples are the Boganiidae-Boganiinae (perhaps related to the Kara Tau *Parandrexis* Martynov)…’ Later, Crowson ([58]: 99) stated: ‘I have seen from the supposedly late Jurassic Kara Tau deposits of Soviet Central Asia, more or less similar to *Parandrexis* Martynov (1926) show some similarities to Boganiidae, but have longer antennae with no trace of a club (weak in *Paracucujus*, etc.)’. The description of the family Parandrexidae with the morphological and bionomical argumentation based on resemblance and probable relationship was published [59] with the list of structural parallel transformations in anthophagous nitidulids and parandrexids, and with a supposition that the latter could be associated with ‘microstrobiles of such Mesozoic gymnosperms as the cheirolepids, bennettitales, or sago palms’ ([59]: 72). Later, Kirejtshuk et al. [60] erroneously supposed that Parandrexidae could have some relation to cleroids rather than cucujoids. Further studies of this fossil family and also of fossil nitidulids, kateretids, smicripids, apophisandrids and the described holotype of the cretoparacucujine surrounded by pollen grains [25] supported the Kirejtshuk’s initial concept on relation of these familes as well as Crowson’s ideas on the relationship between the parandexids and boganiids. Many similarities are also expressed in the characteristics of apophisandrids and parandrexids, particularly in the structure of the head, with the very peculiar mouthparts and antennae, metaventrite, tibiae, tarsi and ovipositor (see the Taxonomic notes below).

The circumstances considered above support the hypothesis that the ‘nitidulid’ group of families should include the families Apophisandridae, Kateretidae, Nitidulidae, Parandrexidae and Smicripidae, which can be regarded as consisting of two lineages rather distinct in the genital structures of both sexes: one of these lineages is represented by the family Nitidulidae alone, while other families of the ‘nitidulid’ group of families form another phyletic lineage (see above). The multiple parallelisms in structural modifications of various organs in the ‘nitidulid’ group of families can be treated as an evident support of the close relation of these families. In addition, the family Nitidulidae can be treated as a couple of lineages (carpophiline and nitiduline lineages—see above), while the small nitidulid subfamilies are represented by one genus, Calonecrinae and Maynipeplinae, and also, it could be more reasonable to consider the numerous subfamily Cybocephalinae as separate families. In this case, the system of this group of families seems to become more balanced, but these changes need some more comprehension of the already available and new data, particularly the examination of new fossils awaiting description.

The parandrexid subfamily Cretoparacucujinae subfam. nov., whose type genus was initially proposed within members of the Boganiidae [25], based mostly on the similarity of mouthparts and legs, and including the ‘head frons with median endocarina and mandibles with a dorsal setose cavity near the base, are distinctive among Cucujoidea.’ As to the dorsal setose cavity on the upper surface of the mandibles, Crowson in 1995 ([4]: 78) held that ‘mandible pockets of Meligethinae or Boganiidae could function as mycangia’ and also stated the following: ‘In Boganiidae (of which Boganiinae have been reported to breed in flowers of Angiospermae, while Paracucujinae, as far as known, breed in male cones of cycads) the dorsal mandible pockets are highly developed and may contain pollen grains. There are few, if any, other characteristics suggesting a direct relation of Meligethinae (or other Nitidulidae) to Boganiidae’. Summarizing the data on these pockets (cavities), Escalona et al. ([61]: 1) wrote the following: ‘Dorsal setose mandibular cavities also occur in Sphindidae and glabrous cavities in a similar position are known in some Phloeostichidae and Silvanidae; all of these taxa, however, feed on fungi or Myxomycetes and may utilize these structures for carrying spores.’ According to the personal observations of the senior author of this paper, these cavities (pockets) in mandibles among different meligethines are rather variable and apparently somewhat congruent with the lateral angles of the frons; i.e., their appearance could have originated as a sequence of structural interactions of the frons and mandibles rather than a function of transmission of spores or pollen. Despite the possible morphogenetic origin of these mandible pockets, they can be also used to transport spores and pollen. Thus, the structural features of the mandibles above considered can scarcely give a reason to use them as a direct trace of the phyletic interconnections, although they can represent mediated evidence of these interconnections with homoplastic appearance. The ‘median endocarina’ (midcranial suture) is often observable in Apophisandridae **stat. nov.** and Parandrixidae, which also supports the approaching of parandrexinds and apophisandrids by Vitali [56]. Oppositely, the midcranial suture in representatives of the abovementioned groups apparently could be interpreted as their plesiotypic character.

The relationships between the ‘nitidulid’ group of families and other cucujoid families in a wide sense (but not split into many superfamilies as was recently published [10]) are differently interpreted in many systematic and phylogenetic models (e.g., [10]), but there needs to further comprehensive analysis of their characteristics, appearance and presence, which might be connected with circumstances beyond the matrixes compiled from the structural characteristics defined by researchers. A great level of the structural similarity of the family Boganiidae with the families Apophisandridae and Parandrexidae could be conditioned not only by adaptations to a similar lifestyle, but also somehow predetermined by a joint phyletic root. The Boganiidae in particular is very probably a group closely related to both apophisandrids and parandrexids, taking into consideration the adult structures of mouthparts and appearance of its imaginal pollinophagy on gymnosperms and anthophagy on angiosperms (see also the Notes and Diagnosis of Apophisandridae stat. nov. below). Although Crowson regarded boganiids as a related group to nitidulids and/or parandrexids (see above), there are many other interpretations of the relationships regarding this family [61,62]. It seems quite acceptable that structural parallelisms themselves are more likely in the groups with closer relationships. This hypothesis should be more grounded by further studies of fossils of all these families. It may be that the families Helotidae and Nitidulidae could be linked based on their structure of the male genitalia and many external structural characteristics, although the helotids seems to maintain too many plesiomorphic structural features [5], and therefore, their relationship with other cucujoids and particularly with nitidulids is a rather delicate and complicated matter [5]. Similarly, the family Monotomidae, with an asymmetric aedeagus, could be also related to the ‘nitidulid’ group of families, and this assumption could be admitted after further serious studies and analysis. The pair of the families Sphindidae and Protocucujidae, as well as other above hypotheses on the interconnections between the ‘nitidulid’ group of families, should also be justified by further studies. These definitions determine the temporary disuse of the name Nitiduloidea *sensu lato* recently proposed [10] and even Nitiduloidea *sensu stricto* with a taxonomic meaning for the here called the ‘Nitidulidae’ group of families (i.e., Apophisandridae, Kateretidae, Nitidulidae, Parandrexidae and Smicripidae). This is possible when, in approaching the analysis of morphological characteristics, sufficient data are found for other important aspects of evolution and phylogenesis that can be made possible by the application of multiple parallelisms and avoid information noise appearing from subjective or erroneous calibrations [39].

### On Pollinophagy and Pollination with Participation of Representatives from the ‘Nitidulid’ Group of Families

The interconnections between insects and plants are always regarded as one of main circumstances determining the evolution and phylogeny of both groups of organisms, and the evolution of the global terrestrial biota since it appeared. The origin of beetles is connected with interconnections of archaic holometabolans with wood in the forests of the Palaeozoic era. Although associations of ancient Mesozoic beetles with generative organs of plants were also admitted in many old publications, Crowson was one of the first to try to interpret real facts in fossils as indirect evidence of interactions of beetles with plant generative organs. He supposed that some Jurassic plant groups could have ‘cantharophily’ and ‘abundant pollen production, general in anemophilous flowers, such as most Gymnospermae, would provide a good food and attractant for adult beetles’ ([8]: 599). Among other things, Crowson thought ([8]: 666) that *Parandrexis*, from the ‘nitidulid’ group of families, could be one of the pollen-eating beetle groups in the Jurassic. Later, *Lebanoretes andelmani* Kirejtshuk and Azar, 2008 [63] from Kateretidae was described from the Lower Cretaceous Lebanese amber, having all diagnostic characteristics as found in other true kateretids and somewhat similar to the species of *Anthonaeus* Horn, 1879 [64], but with elytra leaving the last three abdominal segments uncovered (but not one-two last abdominal segments as in other kateretids). All modern members of this family with known bionomy have anthophagous adults and anthophagous or carpophagous larvae, feeding on angiosperm flowers (both Magnoliopsida or Dicotylodones, and Liliopsida or Monocotyledones). Recently more than ten descriptions of apophisandrids were published from the Albian/Cenomanian Burmese amber, one species of which has syninclusions in the same piece of amber remains of probable gymnosperm Gnetales [27], while another species contains pollen of the eudicot plant of the genus *Tricolpopollenites* (angiosperm) [29]. In addition to the above records, the apophisandrid specimen from the graphical abstract to this paper (with registration number ‘BUB4385′ in the collection of P. Müller, Munich, Germany) has many pollen grains around its mouth and other places of its body integument (see the graphical abstract). Finally, a few years ago, a parandrexid (*Cretoparacucujus cycadophilus*) was described as a boganiid also from the Albian/Cenomanian Burmese amber piece with syninclusions of ‘pollen grains … close to some species of *Cycas*, *Macrozamia*, and *Zamia*…’ [25].

In contrast to some hypotheses developed by Peris with co-authors [65], it is thought that the Crowson’s concept on beetle associations with ancient generative plant organs has a better fit with facts in the fossil record; this is particularly the case with the finding of a true kateretid in the early Cretaceous [63], which is rather similar to the modern kateretids with imaginal anthophagy and larval antho-/carpophagy, and many findings of various Jurassic and Cretaceous parandrexids [59,66,67] and Albian/Cenomanian apophisandrids (see above), bearing witness to a considerable specialization of the Jurassic and early Cretaceous pollinophagous beetles. The number of genera and species of both last groups discovered in the same resources (Jurassic Daohugou and Karatau, Cretaceous Jiulongshan and Albian/Cenomanian Burmese amber) may indicate significant specialization of them at the time of their fossilization. The last circumstance also does not correspond with the opinion that the pollinophagous beetles known from the Middle Mesozoic were not specialized [65]. This means that the co-evolutionary parallelisms between ancient pollinophagous beetles and their host plants can be admitted even in the Middle Jurassic. Similarly, traces of such co-evolutionary parallelisms among the modern nitidulids and kateretids are demonstrated in a considerable coincidence between both closely related groups of beetles and their host plants ([1,68,69], etc.).

The poorly understood but probably rather important circumstance is that pollinophagous insects could be necessary for normal plant reproduction, even for the anemophilous plants with dioeciousness, which seemed to be characteristic of many Mesozoic plant groups. One of the expressive examples of proof of the latter is the case of the attempts to introduce the dioecious African palm *Elaeis guineense* to South and Central America to obtain its oil without African pollinophagous beetles (epuaraeines and meligethines) living only on its male inflorescences (e.g., [70], etc.). It became successful after the hybridization of African palm with American congener *Elaeis oleifera* and the actions of local American pollinophagous beetles from Mystropini (Nitidulinae) on this plant hybrid ([69,71], etc.). At the same time, beetles from other groups (not nitidulids) live on the female inflorescences of these palms, the larval development of which may take place in these inflorescences (larval carpophagy). Thus, it is thought that some interdependence between anemophilous plants and pollinophagous beetles with small body visiting male cones/inflorescences could have a somehow obligatory character important for anemophilous plants, including for productivity of the latters. Such experiments and many others are impossible for fossils, and therefore, the criticism of the problem of participation of fossil beetle species in such interactions is quite reasonable [72]. Nevertheless, the similar structural adaptations of apophisandrids and parandrexids in mouthparts present an essential mediated support for such a hypothetical conclusion. Another conclusion, one which is possible to draw by taking into consideration the scope of the general structural variability in both abovementioned families, is that only some apophisandrids and probably all known parandrexids could really be associated with actual ancient pollinophagy. Similarly, it can be compared with different unrelated modern groups of Epuraeini or different groups of the nitidulin *Aethina* complex of genera and many other nitidulids, some of which transited to regular anthophagy even without expressing structural modifications. This is another essential circumstance in the life of small-sized pollinophagous beetles, as many nitidulids, can be rather specialized in preference of their food, i.e., plants. Adult nitidulids also usually add soft parts of inflorescences to their pollen diet. Thus, some apophisandrids could be only partly pollinophagous even without identifying structural evidence in mouthparts and antennae as, for example, that of the apophisandrid described as a nitidulid [35]. The personal observation of the midgut of many different modern representatives of the various subgenera of the genus *Meligethes* Stephens, 1830 [73], in the traditional sense [1] but not divided into many ungrounded ‘genera’ [68], showed that the pollen grains can pass through the beetle digestive system almost unbroken but only cracked for the penetration of digestive juice inside grains. The latter circumstance explains why many small anthophagous beetles have chewing mandibles without special structural modification, which is enough to press and crack pollen grains. Many of their other imaginal organs also have no structural specialization for anthophagy. Therefore, it can be supposed that if some anthophagous epuraeines, meligethines or nitidulines were found in fossils by researchers without knowledge of their diet and lifestyle in modern probable descendants, they would scarcely recognize their anthophagy at all ([15,20], etc.).

The paleontological data recently obtained make it possible to estimate them in accordance with the relationships between the ‘nitidulid’ group of families, including the regular parallel trophic changes in ways of phyllophagization, summarized in the scheme in Figure 5 for the family Nitidulidae (for more details, see [38,74], etc.). There are no known cases of a reliable association of modern or fossil nitidulids with cycad cones or gymnosperms. Among nitidulid adults collected on cycads, only Australian *Brachypeplus barronensis* Blackburn 1902 [75] occurs on *Macrozamia communis*; however, it is still not known whether the visiting of male cones by adult beetles is associated with the development of their larvae in these cones. It is also thought that adults of this beetle species can be only visitors of these cones, as it is likely that other cillaeines are rather frequently collected on different angiosperm flowers, particularly on decaying inflorescences. Furthermore, the subfamily Cillaeinae, taking into consideration the pattern of the global cillaeine distribution and the composition of this subfamily in the Recent fauna of the Australian Region, seems to have come to Australia not so long ago and, therefore, cannot have a very old association with Australian cycads. Other modern anthophilous nitidulids are associated with the generative plant organs of angiosperms. Many modern nitidulid groups are characterized by an obligatory association with angiosperm flowers, corresponding with the IId stage of the trophic transitions to phyllophagy (different groups of Epuraeinae, Carpophilinae, Nitidulinae (Nitidulini, Cychramini and Cyllodini), and Cillaeinae), while only a few groups independently reached the IIId stage (*Nitops* Murray, 1864 [76] from Carpophilinae, all Meligethinae, all nitiduline Mystropini and some genera of the *Aethina* complex of genera (Nitidulini) and Cyllodini). The IIIth stage is also known among Carpophilinae (complete carpophagy in different subgenera of *Carpophilus* Leach in Stephens, 1830 [77] and subgenus *Anophorus* Kirejtshuk, 1990 [78] of the genus *Urophorus* Murray, 1864 [76], etc.), and the IVth stage with imaginal anthophagy/phyllophagy and larval phyllophagy is known, for example, in nitiduline Cychramini (*Xenostrongylus* Wollaston, 1854 [79]) and Cyllodidi (in particular, probably all representatives of *Camptodes* Erichson, 1843 [23]). In addition, adults of many nitidulid groups known as regular visitors of angiosperm inflorescences have unstudied larvae, and therefore, their position in the above scheme is not yet defined.

As all the modern kateretids with known bionomy are characterized by the imaginal anthophagy and larval anthophagy/carpophagy, so it can be expected that the same could pertain to fossil representatives with similar appearance and structural features (*Heterhelus* and *Eoceniretes* Kirejtshuk and Nel, 2008 [12] could have the same characteristics, while *Lebanoretes* and *Antirhelus* gen. nov., with some structural differences, could have somewhat different diets in both adults and larvae). The modern smicripids with a known bionomy have anthophagous adults and mycetophagous larvae (IId stage of throphic changes according to the above considered scheme). The fossil members of this family, with very similar appearance and structural features, seemed to have the same trophic characteristics. As to Parandrexidae and Apophisandridae stat. nov., it can be admitted with a great probability that they had imaginal pollinophagy, could be, on both gymnosperms and angiosperms, for all genera of the former because of their uniform set of characteristics, and for a part of the latter because of their rather considerable imaginal structural variability.

## 5. Taxonomic Notes

(A)**Family Apophisandridae** Molino-Olmedo, 2017 [34], **stat. nov.** and its position
-Apophisandrini Molino-Olmedo, 2017 [34]; Vitali, 2019 [56]

Type genus *Apophisandra ammytae* Molino-Olmedo, 2017 [34], fossil, Burmese amber, Cretaceous, Albian/Cenomanian.

**Notes**. This family is rather numerous and diverse, although only a small part of it has been already described. It is like that after a further revision, it will be split into some subfamilies. One of main aspects differentiating it from Kateretidae is the structure of the tarsi (Figure 3B,D; graphical abstract to this paper and ([29]: Figure 1e,f), which are greatly different in structure of the first four tarsomeres: the apophisandrids have unilobed all four tarsomeres (sometimes tarsomere 4 is much smaller than previous ones), while the kateretids have the first three tarsomeres bilobed and a subcylindrical tarsomere 4. At the same time, the variability of apophisandrid tarsi are so great that sometimes each of the tarsomeres 1–4 looks almost subcylindrical. However, in most cases, when tarsi are clearly visible, the lower surface of tarsomeres 1–4 is longer than their upper surface. In contrast to all described apophisandrids, the genus *Scaporetes* is medially convex, some undescribed genera also have a comparatively convex upper surface, but the pronotal and elytral sides in these cases are more or less explanate (not narrowly explanate, as in kateretids). The apophisandrid genus *Protonitidula* was proposed as a nitidulid one, perhaps, because both of these families are rather variable in many structural characteristics as a result of the parallel structural transformations of many organs; also, the habitus of the abovementioned fossil is somewhat reminiscent of some epuraeins, but its scapus, its rather loose and not dorsoventrally compressed antennal club, its deeply incised posterior edge of the metaventrite between coxae and also its particularly serrate pygidial apex, tibiae and tarsi are different from those in nitidulids. In addition, this specimen has a weak sclerotization of integument, making it possible to clearly observe the outline of the ovipositor ([35]: Figure 1), which is similar to that in kateretids and parandrexids, but not that in members of any subfamily of nitidulids (see above).

Finally, it is preliminarily thought that the families Apophisandridae **stat nov.**, Boganiidae and Parandrexidae could be so closely related that it is reasonable to treat them as subfamilies of the same family, which can be somehow comparable with the family Nitidulidae, although, on the other hand, splitting of the Nitidulidae in current interpretation could give even a more reasonable systematic construction. However, to recognize one of these concepts, it is necessary to increase our knowledge on fossils of the mentioned groups and to reach a comparable level of understanding of each one. This supposition also corresponds with the ideas of Crowson and [8,58] and Vitaly [56]. The important step for testing of this is the finding of *Palaeoboganium jurassicum* Liu, Ślipiǹski, Lawrence, Ren and Pang, 2017 from the Middle Jurassic of Daohugou [61], with its body size comparable with the Jurassic parandrexids [67] and having some resemblance to the modern members of the subfamily Paracucujinae, whose larvae and adults feed on male cones of *Macrozamia riedlei* (complete pollinophagy/anthophagy—Figure 5).
Figure 5Scheme of ways of changes in trophic associations in the family Nitidulidae to appearance of phyllophagy (after Kirejtshuk, 1989 [38], translated with some correction). The Latin numerals (I, II, III, IV) are for designation of stages of change from initial complete (xylo)phagy.
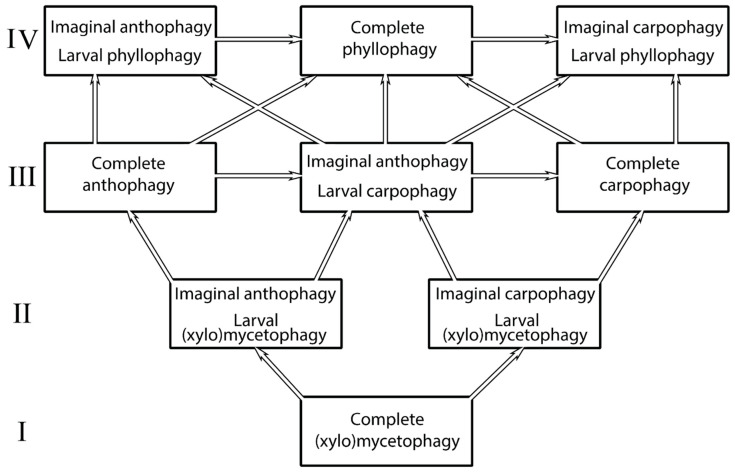



**Diagnosis** (taking into consideration both described and undescribed species—see the graphical abstract to this paper and Figure 3B–D). Body usually comparatively small to medium-sized (1.7–6.0 mm), oval or elongate oval, subflattened or slightly to moderately convex dorsally and slightly convex ventrally (except *Scaporetes* with medially rather convex pronotum and elytra); upper integument usually shining to rather mat; dorsal pubescence usually rather fine and uniform; cilia along pronotal and elytral sides usually well raised (sometimes very long) to almost inconspicuous. Body integument with sparse diffuse punctures and smoothed interspaces (sometimes punctation rather dense and coarse, interspaces with relief and dense microreticulation) or sometimes elytra with seriate punctures.

Head subflattened dorsally, usually comparable wide with prothorax, depressed at level of moderately raised and laterally prominent eyes, usually with very short frons and usually traced midcranial suture, very short genal processes, antennal insertions slightly covered by dilatations of frons, without expressed antennal grooves; labrum well- and usually far-anteriorly exposed, transversely truncate, deeply and widely excised or with narrow median excision; mandibles frequently with long to extremely long and sharp apices (particularly in males) with more or less raised teeth along inner edge; mentum different in shape; palpi of usual shape or not infrequently with more or less elongate maxillary and sometimes also labial palpi. Antennae of usual shape, although frequently rather long and with particularly modified scape; three-segmented club always loose (not compact) and not compressed, sometimes modified as loosely pectinate.

Pronotum subflattened and at most only slightly narrowing anteriorly; anterior and posterior edges transverse or slightly curved (shallowly bi-emarginate or convex); sides narrowly to widely explanate. Scutellum moderately to usually very large, subtriangular or subpentangular. Elytra transversely abrupted, and remaining 1–3 abdominal segments uncovered, sometimes moderately or strongly costate, sides somewhat explanate; epipleura narrow to rather wide. Uncovered abdominal segments; pygidium various in shape and usually with more or less serrate posterior edge.

Prosternum moderately long, and its intercoxal process with simple apex not expanded laterally at apex. Procoxae are transverse, open posteriorly. Distance between coxae in each pair more or less small (with greatest distance between mesocoxae and sometimes (sub)contiguous between metacoxae). Mesoventrite moderately long and weakly convex. Mesocoxal cavities oval. Metaventrite moderately long, with more or less expressed discrimen, posterior edge between metacoxae (if not (sub)contiguous) rather deeply and narrowly incised. Metepisternum of usual shape. Abdominal ventrites 1–4 of usual shape (ventrite 1 usually longest, or ventrites 1 and 4 comparatively longer than each of ventrites 2 and 3); hypopygidium longer than each of ventrites 2–4.

Legs usually rather long and narrow. Femora of usual shape, moderately wide and with usual variability. Tibiae of usual shape and usually very narrow, with one outer border bearing one row of more or less thick short to rather thick and very long spines; another outer border of meso- and metatibiae somewhat dislodged along upper surface and without raised spines. Five-segmented tarsi rather long, with tarsomeres 1–4 usually lobed to sometimes nearly simple (but not bilobed, and tarsomere 4 usually not subcylindrical and very short); tarsal claws frequently strongly toothed to bulged at base or simple.

Male anal sclerite moderately to greatly raised and always very largely exposed and frequently serrate along posterior edge. Aedeagus apparently with articulated paramera ([26]: Figure 4A). Ovipositor with more or less weakly sclerotized (nearly membranous) gonocoxites, without clearly separated inner and outer lobes and moderately raised styli ([35]: Figure 1).

Sexual dimorphism, in addition to structure of ultimate abdominal segment, with presence of anal sclerite and genitalia, usually more or less expressed in shape of head and prothorax, structure of antennae, particularly in basal antennomeres, and also in labrum, mandibles (sometimes with very long and acuminate apices) and palpi.

**Comparison**. In contrast to Apophisandridae stat. nov., Kateretidae have their body, in general, rather or moderately convex dorsally (modern *Anthonaeus* and *Platamartus* Reitter, 1892 [80], and also fossil *Lebanoretes* with moderately convex dorsum); pronotal and elytral sides without distinct cilia; head usually flattened dorsaly and moderately projecting anteriorly, without depression on frons (except *Antirhelus* gen. nov.), never with midcranial suture, labrum is usually without median excision, mandibles are moderately developed and without long and sharp apices, palpi are of usual shape and moderately short; antennae with compact two–three-segmented subcylindrical club (not very loose); pronotum, as a rule, is strongly convex at disc, narrowly explanate or not explanate or at most only widely (sub)explanate at sides, elytra are strongly convex (only sometimes moderately convex) and with scarcely visible sides from above), pygidium of various shape but never with serrate posterior edge; mesal prohypomeran process is joined but apparently not fused with the lateral dilatations of the prosternal process; distance between metacoxae is markedly greater than that between pro- and mesocoxae; posterior edge between metacoxae is shallowly emarginate to nearly rectilinear (never deeply excised); abdominal ventrite 4 is usually much wider than the visible base of hypopygidium; all tibiae are without both distinct outer border(s) and row of spines, tarsomeres 1–3 are bilobed, and tarsomere 4 is subcylindrical and smallest (slightly exposed from lobes of the tarsomere 3).

The family Apophisandridae stat. nov. differs from Parandrexidae in their shorter and wider body (not elongate as in parandrexids), head is not larger and usually not longer than prothorax, moderately sized and laterally prominent eyes, eyes only sometimes located at anterior edge of head (head not strongly larger and longer than prothorax, and eyes never very small, and their outer edge does not form one line slightly and gently curved together with the outlines of frons and temple as in parandrexids, although eyes of cretoparacucujines are of usual size and moderately prominent), antennae are moderately long to sometimes very long, always clubbed (antennae are not very long, subfiliform and without a trace of club as in parandrexids), pronotum is of usual shape and wider than head, slightly to moderately convex, in posterior half moderately narrowing posteriorly (pronotum is not narrower than head, subdepressed and narrowing posteriorly from its anterior half as in parandrexids), scutellum comparatively large and subangular at apex (not very small and transverse at apex as in parandrexids), elytra are rather shortened and truncate at apices (but not complete or almost complete elytra conjointly arcuate at apices as in parandrexids), pygidial apex is frequently serrate (never serrate as in parandrexids), male anal sclerite is well exposed and frequently serrate at its posterior edge (not scarcely exposed as in parandrexids), femora are moderately wide, tibiae with expressed outer border and row(s) of spines (not narrow femora and tibiae, and tibiae are also without both expressed border and row of spines as in parandrexids).

The family Apophisandridae stat. nov. differs from Smicripidae in the frequently elongate ultimate labial and maxillary palpomeres (ultimate labial palpomeres not cup-shaped as in smicripids), usually much looser antennal club, gently sloping and (sub)explanate pronotal and elytral sides, posterior edge of metaventrite narrowly and deeply excised (not very great and shallowly emarginate as in smicripids), all tibiae with rows of more or less raised spines along outer edge (not protibia crenelate along outer edge, and meso- and metatibiae without setose borders along outer edge as in smicripids), unilobed tarsomeres 1–4 (tarsomeres 1–3 not bilobed, and antennomere 4 not subcylindrical as in smicripids), and usually serrate pygidial posterior edge.

The family Apophisandridae stat. nov., in terms of the level of variability of many structures, is somewhat similar to Nitidulidae, which is, however, known to have much greater number of members and wider scope of variability in many structures in representatives of both recent and fossil faunas. The strict indication on the distinctness of both groups represents the genitalia of their both sexes: paramera apparently articulated with phallobase in apophisandrids ([26]: Figure 4A) and tegmen in nitidulids; and almost evenly membranous ovipositor without clearly separated inner and outer lobes of gonocoxites in apophisandrids ([35]: Figure 1), while the nitidulid ovipositor usually has the traced sclerotized elements and separated inner and outer lobes of the gonocoxites. The more or less good diagnostic difference in tarsi (nitidulid tarsomeres 1–3 are bilobed, and tarsomere 4 is very small and subcylindrical), although nitidulids almost never have unilobed tarsomeres 1–4, but sometimes, all tarsomeres become simple (subcylindrical) and long, and even sometimes look almost unilobed (e.g., tarsomeres 1–3 in some cybocephalines). The nitidulid posterior edge of the metaventrite between coxae are usually nearly rectilinear or shallowly arcuately emarginate (but not deeply and narrowly excised as in apophisandrids). In contrast to apophisandrids, nitidulids usually have a crenelate outer edge of protibia and two or one border(s), each of them with row of fine setae. Besides, if only some nitidulid cillaeines have the serrate posterior edge of female pygidium and male anal sclerite, while all apophisandrids are characterized by the distinctly serrate posterior edge of these sclerites in the most cases.

Finally, Apophisandridae stat. nov. has a considerable similarity to some Boganiidae in the anterior part of the head and characteristics of the exposed mouthparts, absence of antennal grooves, procoxal cavities open posteriorly, very short and narrow prosternal process without lateral dilatations at apex, posterior edge of metaventrite, and unilobed tarsomeres 1–4, but it is distinct from the latter in the slightly to moderately convex body with gently sloping pronotal and elytral explanate sides (not steeply sloping and at most very narrowly explanate pronotal and elytral sides as in boganiids), clubbed antennae (although some boganiids show a tendency to form a club), scutellum comparatively large and subangular at apex (not very small and transverse at apex as in boganiids), truncately abrupt elytra leaving 2–3 last abdominal segments uncovered, usually a crenelate posterior edge of pygidium, and strongly exposed male anal sclerite.

**Composition**. Type genus (*Apophisandra* Molino-Olmedo, 2017 with *A. ammytae* Molino-Olmedo, 2017 [34]) and eight other genera: *Cretaretes* Peris and Jelínek, 2020 with *C. minimus* Peris and Jelínek, 2020 [28], *Electrumeretes* Peris and Jelínek, 2020 with *E. birmanicus* Peris and Jelínek, 2020 [27], *Furcalabratum* Poinar and Brown, 2018 with *F. burmanicum* Poinar and Brown, 2018 [26], Pelretes Tihelka, Li, Fu, Su, Huang and Cai, 2021 with *P. bicolor* Zhao, Tihelka, Huang and Cai, 202 [30] and *P. vivificus* Tihelka, Li, Fu, Su, Huang and Cai, 2021 [29], Polliniretes Peris and Jelínek, 2020 with *P. penalveri* Peris and Jelínek, 2020 [27], Protokateretes Zhao, Huang and Cai, 2023 with P. antiquus (Peris and Jelínek, 2020) ([30]: (Eoceniretes), [32]) and *P. megalocephalus* Zhao, Huang and Cai, 2023 [30], Protonitidula Zhao, Huang and Cai 2022 with P. neli Zhao, Huang and Cai 2022 [35] and *Scaporetes* Zhao, Huang and Cai 2023 with S. rectus Zhao, Huang and Cai 2023 [32]. All mentioned genera were proposed for the species originated from the Albian/Cenomanian Burmese amber.

**Notes on probable bionomy**. A resemblance of some fossil apophisandrid adults with modern kateretids and the occurrence of the former in amber pieces with pollen make it possible to admit a pollinophagous diet, at least for some apophisandrid adults. The considerable number of probable species and genera of this family in fossils also allows us to suppose a rather great level of trophic specialization and variability in habitats in its different groups. Some apophisandrids with generalized appearance could be associated with fungi (see the Discussion section above).

(B)Fossil records of the **family Kateretidae** Kirby, 1837 [81]

**Composition of fossil members**. Only two paleoendemic genera are known in this family, *Lebanoretes* Kirejtshuk and Azar, 2008 [63] from the early Cretaceous Lebanese amber (*Lebanoretes andelmani* Kirejtshuk and Azar, 2008) and *Eoceniretes* Kirejtshuk and Nel, 2008 [12] from the earliest Eocene Oise amber *Eoceniretes yantaricus* Kirejtshuk and Nel, 2008 [12]. The *Heterhelus* species is also found in Oise amber (*Heterhelus expressus* Kirejtshuk and Nel, 2008) [12]. The genera *Kateretes* Herbst, 1793, recorded from the Eocene Baltic amber (Hope, 1836 [82]), and *Amartus* LeConte, 1861 [83], represented by *A. petrefactus* Wickham, 1912 [84], were described from the late Eocene of Florissant Fossil Beds (Colorado, USA). Both last mentions should be regarded as doubtful, because the first was not documented at all, but the second was provided with a rather obscure description and drawings, which do not make certain the family attribution of the described print (although the general outline of the body and particularly the antenna are very reminiscent of those in kateretids—according to the descriptor of it, the holotype of *Amartus petrefactus* had, at the time of description, a rather good condition and fitted with the modern Nearctic members of *Amartus*).

**Notes on probable bionomy**. If the characteristic bionomy and trophic specialization of the modern kateretids ([3,85], etc.) can be extrapolated in the past, the Caenozoic kateretids could be completely anthophilous beetles with imaginal anthophagy and larval anthophagy/carpophagy, while the early Cretaceous *Lebanoretes* could be associated with gymnosperm plants.

(C)**Subfamily Antirhelinae** Kirejtshuk, **subfam. nov.**

Type genus *Antirhelus* gen. nov.; fossil, Rovno amber, Eocene.


http://zoobank.org/urn:lsid:zoobank.org:act:5AF84A5C-9EAB-48B2-B62A-E32DAEFA54D3


**Notes**. The new subfamily is proposed because of the extremely peculiar head structure in the type species of *Antirhelus* gen. nov., very different from that in all known kateretid genera and even from that of all anthophilous modern representatives from the ‘nitidulid’ group of families, although the short frons before eyes is rather characteristic of fossil apophisandrids and parandrexids. The frons of all other kateretids are flat or at most slightly depressed and very slightly declined before the exposed labrum, without any clear structural formation. At the same time, other structural features of *Antirhelus* gen. nov. seem to fit separate ones in different kateretine genera (see the Diagnosis of the new genus below).

**Diagnosis**. Head short and slightly projecting anteriorly and sharply declined downwards before antennal insertions, frons conjointly covering by dilatation over antennal insertions and Y-formed depression before transverse and moderately exposed labrum.

**Comparison**. See the Notes to this subfamily, above, and Diagnosis to the new genus, below.

(D)**Genus *Antirhelus*** Kirejtshuk, **gen. nov.**

Type species *Heterhelus buzina* Kupryjanowicz, Lyubarsky and Perkovsky, 2021 [57]; fossil, Rovno amber, Eocene.


http://zoobank.org/urn:lsid:zoobank.org:act:46F136EF-72D0-428A-8EBF-2BC6B9CF1760


**Etymology**. The name of this new genus is formed from the Greek prefix ‘αντί’ used with other words to mean opposite or opposition and the root of the generic names among Kateretidae ‘*rhelus*’ (‘*Heterhelus*’ and ‘*Sibirhelus*’). Gender masculine.

**Diagnosis** ([57]: Figures 1 and 2). Body moderately small (circa 1.8 mm), elongate and subparallel sided, rather convex dorsally and moderately ventrally, covered with short and moderately dense vestiture. Head short and slightly projecting anteriorly before moderately raised and moderately prominent eyes, sharply declined downwards before antennal insertions, frons conjointly covering by dilatation both antennal insertions and Y-formed depression before transverse and moderately exposed labrum, mandibles comparatively small and slightly exposed, mentum subpentangular and more than twice as wide as long. Antennae 11-segmented, scape and pedicel swollen (somewhat similar to those in *Kateretes pedicularius* (Linnaeus, 1758) and *K. pusillus* (Thunberg, 1794)): scape along whole length and pedicel mostly at apex), three-segmented club compact and elongate, not compressed. Pronotum very convex, arcuate and extremely narrowly explanate at sides, subtruncate at anterior and posterior edge, posterior angles almost distinct and very slightly projecting posteriorly. Scutellum rather large, transversely subtriangular and subangular at apex. Elytra about one and half as long as wide together, narrowly explanate at sides and forming widely open sutural angle. Two last abdominal segments uncovered by elytra. Abdominal ventrite 4 about as long as ventrites 2 and 3 together. Subtransverse apex of male anal sclerite very slightly exposed from under abdominal segment VII. Legs of usual structure, all tibiae gradually and slightly widening distally. Tarsi 5-5-5, with tarsomeres 1–3 rather bilobed and tarsal claws simple.

**Comparison**. The new genus is distinct from all known kateretid genera in the head structure (see the Diagnosis section above). Each of its other characteristics demonstrate considerable similarity with those of separate kateretine genera. The most expressed characteristic seems to be in the shape of the antennomeres 1 and 2, which can be compared only with those in members of *Kateretes* (see the Diagnosis above). *Antirhelus buzina* comb. nov. also differs from *Kateretes* in the almost distinct top of posterior angles of pronotum. The new genus is reminiscent of some katertine genera with the more or less distinct posterior angles of the pronotum differing from them in the following characteristics:-From *Brachypterolus* Grouvelle, 1912 [86]: also in the shape of the antennomeres 1 and 2, not strongly projecting posterior angles and not strongly bisinuate posterior edge of the prothorax, much shorter penultimate abdominal segment, and simple tarsal claws;-From *Heterhelus*: also in the shape of antennomeres 1 and 2, and almost distinct posterior angles of the pronotum (species of *Heterhelus* with the posterior angles of the pronotum usually almost or distinctly rounded);-From *Lebanoretes*: also in the more convex body, shape of antennomeres 1 and 2, and shorter abdominal segments with only two segments exposed from under elytral apices;-From *Sibirhelus* Kirejtshuk, 1989 [87]: in the shape of antennomeres 1 and 2, distinct antennal club, markedly shorter prothorax, much shorter and not curved protibia, and simple tarsal claws.

**Composition**. The type species only.

**Notes on probable bionomy**. Taking into consideration the structure of head in the type species (‘*Heterhelus*’ *buzina*), its feeding on inflorescences of *Sambucus* (in Russian ‘бyзинa’—‘buzina’) characteristic of modern *Heterhelus* species with known bionomy can be scarcely recognized.

(E)**Family Parandrexidae** Kirejtshuk, 1994 [59] and its composition

**Notes**. This family was proposed for the genus *Parandrexis* alone, published by Martynov in 1926 [88], although a part of further additional research described congeners [67], which could be considered as at least one separate genus remaining unnamed, and also, one more genus (*Martynopsis* Soriano, Kirejtshuk and Delclòs, 2006) has already been published [66]. Crowson 1981 supposed a close relation of *Parandrexis* to modern Boganiidae [8]. New data on fossils make it possible to construct an essential argument in favor of a close relationship between these mentioned families (see the Discussion section above). *Parandrexia beipiaoensis* (Hong 1983 [89]) from the Haifanggou Formation (Middle/Upper Jurassic) of Liaoning (China) was described with many evidently wrong details [59], and therefore, they are omitted in the amended diagnosis below, and its name, *Parandrexia* Hong 1983 [89], is regarded as a junior synonym of *Parandrexis*. However, the description of the Jurassic members of this family from Daohugou [67] brought some important information to amend the initial diagnosis of this family [59].

**Amended diagnosis** (taking into consideration both described and undescribed species—see Figure 4). Body small- to medium-sized (1.8–15.0 mm), elongate, subflattened or at most slightly convex dorsally and ventrally; upper integument seemingly smoothed and apparently mat; dorsal pubescence (if visible) seemingly rather fine and subuniform. Body integument usually with sparse diffuse punctures and obliterated interspaces or sometimes elytral punctation seriate.

Head very large and evenly convex to subflattened dorsally, usually comparable wide with prothorax or markedly wider, evenly subflattened (but not depressed), with extremely short frons before rather small and not prominent eyes (in Cretoparacucujinae subfam. nov. eyes moderately sized and of usual prominence), sometimes with expressed midcranial suture, temples very long and very slightly and gradually narrowing posteriorly, genal processes moderately wide and usually far projecting anteriorly, antennal insertions slightly or widely covered by dilatations of frons, without expressed antennal grooves; labrum well exposed and rather variable in size and shape, sometimes with median deep excision; mandibles frequently with long or very long and sharp apices (particularly in males); mentum different in shape; palpi of usual shape or not infrequently with more or less elongate maxillary and sometimes labial palpi. Antennae moderately long and submoniliform (*Martynopsis*) to usually rather long, subfiliform and consisting of all antennomeres rather elongate.

Pronotum subflattened, usually significantly smaller than head (only in *Parandrexis subtilis* Kirejtshuk, 1994 [59] head shorter than prothorax, probably because of its retraction inside prothorax), usually widening anteriorly (in Cretoparacucujinae subfam. nov. pronotum subquadrangular and with more or less rounded anterior and posterior angles), anterior and posterior edges transverse or very slightly curved; sides widely (sub)explanate. Scutellum rather small, transverse and frequently widening apically, subtruncate to widely rounded at apex. Elytra complete or slightly shortened and conjointly rounded at apices, sometimes striate, epipleura moderately wide. Pygidium of various shapes, apparently rather short and with simple posterior edge (not serrate).

Prosternum comparatively short and its intercoxal process with simple apex, apparently without lateral dilatations at apex. Procoxae transverse, apparently open or not completely closed posteriorly. Distance between coxae in each pair more or less small (or (sub)contiguous between metacoxae). Mesoventrite very short and apparently weakly convex. Mesocoxal cavities oval. Metaventrite moderately long, with more or less raised discrimen, posterior edge between metacoxae (if not (sub)contiguous) rather deeply and angularly incised. Metepisternum of usual shape and proportions. Abdominal ventrites 1–4 of usual shape (ventrite 1 usually longest, or ventrites 1 and 4 comparatively longer than each of ventrites 2 and 3); hypopygidium usually longer than each of ventrites 2–4, and its posterior edge not serrate.

Legs usually rather long and narrow. Femora of usual shape, comparatively narrow and with usual variability. Tibiae comparatively narrow and subparallel sided, of usual shape without expressed border or row of spines. Five-segmented tarsi rather long, tarsomeres 1–4 apparently simple or lobed (but not bilobed, and tarsomere 4 not subcylindrical and very short); tarsal claws simple.

Male anal sclerite at most scarcely exposed. Ovipositor slightly sclerotized and gonocoxites without clear separation of inner and outer lobes (known in Cretoparacucujinae subfam. nov. ([25]: Figure S2H).

Sexual dimorphism usually expressed in length of antennae, length of mandibles, shape and proportions of head and pronotum, and also probably in length of palpi and shape of mentum.

**Comparison**. The general wide comparison was published with the description of this family [59]. See the Diagnosis and Comparison to Apophisandridae stat. nov. above and the Diagnosis and Comparison to the subfamily Cretoparacucujinae subfam. nov. below. This family shares many diagnostic characteristics with Boganiidae, e.g., in the structure of the prosternal process, with procoxal cavities open posteriorly, the shape of the comparatively small scutellum being widened apically and with subtruncate apex, and some others. Nevertheless, Parandrexidae differ from the latter in the usually subflattened body with comparatively widely (sub)explanate pronotal and elytral sides, very large head with more or less small eyes located in the anterior part of the head (but eyes are not located basally and are comparatively large as in Boganiidae), and very close to (sub)conjoining metacoxae. Some structures of both groups demonstrate the tendencies towards the same transformations in modifications of the shape of the head, features of the integument, observable elements of the mouthparts and antennae, and also tarsi.

**Composition**. The subfamily Parandrexinae *sensu stricto* includes the type genus (*Parandrexis*), *Martynopsis* and some other undescribed genera from the Middle Jurassic and Lower Cretaceous, while another subfamily, Cretoparacucujinae subfam. nov., has only one described genus from Albian/Cenomanian Burmese amber and at least two genera from the early Cetaceous of Liaoning (China) waiting description.

**Notes on probable bionomy**. This family was apparently associated with male cones of gymnosperms having at least imaginal pollinophagy.

(F)**Subfamily Cretoparacucujinae** Kirejtshuk, **subfam. nov.**

Type genus *Cretoparacucujus* Cai, Escalona, Li, Huang, and Engel 2018 [25]; fossil, Burmese amber, Cretaceous, Albian/Cenomanian


http://zoobank.org/urn:lsid:zoobank.org:act:99D98C39-8E0C-47F7-973C-01764C6DB319


**Notes**. This new subfamily is proposed based on the description of *Cretoparacucujus cycadophilus* Cai, Escalona, Li, Huang, and Engel 2018 [25] from the Cretaceous Burmese amber (Albian/Cenomanian) and another undescribed species of unnamed genus studied from the Yixian Formation (Lower Cretaceous) of Liaoning (China). Also, two specimens from the latter Lagerstätte belonging to the new paraxendrid subfamily are used for comparison to check some characteristics included in the Diagnosis below. The senior author of this paper had a chance to study specimens from the collection of CNU, where some members of this subfamily were found. One of these specimens is here present in Figure 4B showing the rather long mandibles (only their base was visible), long palpi, modified labrum, midcranial suture, structure of thorax, somewhat shortened elytra remaining uncovered abdominal apex (ultimate and apex of penultimate segments) and other subfamily characteristics. Other specimens of this new subfamily from CNU, in addition, demonstrate the long and curved mandibles and more arcuate pronotal sides, which could give a reason to assign them to a separate genus.

**Diagnosis** (taking into consideration both described and undescribed species—see Figure 4B). Body small to moderately large (1.8–6.5 mm), elongate, subflattened or slightly convex dorsally and ventrally; upper integument seemingly smoothed and mat; dorsal pubescence (if visible) seemingly rather fine, sparse and subuniform. Body integument usually with sparse diffuse and rather fine punctures and obliterated interspaces and without seriation on elytra.

Head very large and subflattened dorsally, comparably wide with prothorax or somewhat longer, evenly subflattened (not depressed), with short frons before moderately small and moderately prominent eyes, temples moderately long to very long, genal processes moderately wide and usually far projecting anteriorly, antennal insertions widely covered by dilatations of frons, without expressed antennal grooves; labrum well exposed and far projecting anteriorly, with median wide or very narrow excision; mandibles with long to very long and sharp apices; mentum trapezium like and moderately sized to rather small; palpi more or less elongate, maxillary ultimate palpomere sometimes very long. Antennae rather long, subfiliform and consisting of rather elongate all antennomeres.

Pronotum subflattened, usually markedly smaller than head, subquadrate with rounded anterior and posterior angles, anterior and posterior edges transverse or very slightly curved; sides slightly to moderately arcuate and widely subexplanate. Scutellum rather small, slightly widened apically, transverse and widely rounded at apex. Elytra complete to somewhat shortened and conjointly rounded at apices, not striate, sides narrowly explanate; epipleura narrow, their apices completely covering abdomen or remaining uncovered elytral apex (1–2 abdominal segments). Pygidium with subtransverse and simple posterior edge, without serration at apex.

Prosternum moderately long, with intercoxal process very narrow and with simple apex. Procoxal cavity transverse, apparently open posteriorly. Distance between pro- and mesocoxae very narrow and that between metacoxae (sub)contiguous. Mesoventrite very short and apparently weakly convex. Mesocoxal cavities oval. Metaventrite moderately long, with more or less expressed discrimen. Metepisternum of usual shape and proportions. Abdominal ventrites 1–4 of usual shape, ventrite 1 longest and at least as long as combined length of ventrites 2 and 3; hypopygidium comparable in length to each of ventrites 2–4.

Legs long and narrow. Femora of usual shape and comparatively narrow. Tibiae of usual shape, rather narrow and apparently without expressed border or row of spines. Five-segmented tarsi rather long, tarsomeres 1–4 apparently simple or lobed (but not bilobed and tarsomere 4 not subcylindrical, very short); tarsal claws apparently simple.

Ovipositor with gonocoxites not clearly divided into inner and lateral lobes and styli located apically.

**Comparison**. The subfamily Cretoparacucujinae subfam. nov. differs from Parandrexinae *sensu stricto* in the more elongate and comparatively smaller body (1.8–6.5 mm), smaller head with eyes more prominent (not forming an even line of the frons, eye and temple as in Parandrexinae *sensu stricto*), antennal insertions widely covered by the dilatations of the frons (not slightly covered as in Parandrexinae *sensu stricto*), subquadrate pronotum with rounded anterior and posterior angles (not pointed anterior angles of the pronotum as in Parandrexinae *sensu stricto*).

This new subfamily differs from Boganiidae in the elongate and subflattened body (not elongate oval and rather convex as in Boganiidae), very large and longer head with eyes located in its anterior half and rather long temples (not transverse with eyes located basally and short temples as in Boganiidae), widely subexplanate pronotal and moderately narrowly explanate elytral sides (not narrowly explanate pronotal and elytral sides as in Boganiidae).

**Composition**. Type genus (*Cretoparacucujus*) from the Albian/Cenomanian Burmese amber and at least two undescribed genera (Figure 4B) from the Lower Cretaceous of Liaoning (China).

**Notes on probable bionomy**. See the Notes on the probable bionomy of the family Parandrexidae above.

(G)Designation of **lectotype** of ***Parandrexis parvula*** Martynov, 1926 [88]

Martynov had for his description two specimens (male and female) from the village of Galkino (East Karatau, Kazakhstan; Middle/Upper Jurassic, Karabastau Formation), collected in 1924 and, at the moment of this study, in the collection of VSEGEI, only the male with the registration number “VSEGEI: 5017-1” is found, which is here designated as the lectotype of *Parandrexis parvula.*

(H)Fossil records of the **family Smicripidae** Horn, 1880 [90]—see Figure 3A

**Composition of fossil members**. The current fossil record includes two genera of this family. One of them is *Smicrips* LeConte, 1878 [90], which is represented by three species recorded in the Cenozoic faunas: *S. europeus* Kirejtshuk and Nel, 2008 [12] from the Lowermost Eocene Oise amber, *S. fudalai* Kupryjanowicz, Lyubarsky and Perkovsky. 2019 [91] from the Eocene Rovno amber and *S. gorskii* Bukejs and Kirejtshuk 2015 [16] from the Eocene Baltic amber. In addition, six species of this genus are known in the recent fauna of Central America). The second genus *Mesosmicrips* Kirejtshuk, 2017 [23] proposed for one species, *M. cretacea* (Cai and Huang, 2016 [25]), from the Albian/Cenomanian Burmese amber and another unnamed species as a potential congener of the latter is shown in Figure 3A (one of syninclusions in the amber piece with the holotype of the new nitidulid species is described here).

**Notes on probable bionomy**. Taking into consideration data on the bionomy of modern representatives of this family and their great similarity with both Cenozoic and Mesozoic ones, imaginal pollinophagy and larval mycetophagy also seem to be very probable for all known fossil members.

## Data Availability

No new data were created or analyzed in this study. Data sharing is not applicable to this article.

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
