# Peer review of "A New Subgenus of the Genus Phenolia (Coleoptera, Nitidulidae) from Myanmar Cretaceous Amber with Taxonomic, Phylogenetic and Bionomic Notes on the ‘Nitidulid’ Group of Families†"

_insects, 2023, doi:10.3390/insects14070647_

Round 1
Reviewer 1 Report
see attached file

Author Response
Many thanks for the positive review of our manuscript. Indeed this paper aims to find a logic way of explanation of phylogeny of the groups closely related to the family Nitidulidae based on real facts and their logic interpretations. And correspondently we wanted to improve the system in correspondence with these interpretations. The reviewer recommends to apply to the phylogenetic interpretation of the system of Coleoptera recently proposed with usage of an integrative paradigm allowing to join facts and ideas without factual base. In our opinion the latter approach presents somehow an alternative approach and therefore the authors did not want to mix these two approaches. The first approach is aimed at understanding phylogeny as a natural process, while the second approach builds phylogeny on the basis of rules that are somehow external to factual processes in nature. Both approaches can be used as complementary. The manuscript was corrected in way to show our approach and improve English in the manuscript.
Reviewer 2 Report
This is an interesting and important for paleontology manuscript made on representative material.
My comments:
Some words in the text are given in a different font size
"The" is not placed before "amber", for example, Baltic amber, Cretaceous amber are without an article.
Justify of male for Phenolia (Palaeoronia) haoranae.
Fransfer descriptions and justifications for Antirhelus gen. nov. for Heterhelus businae) and two subfamily taxa (Antirhelinae subfam. nov. and Cretoparacucujinae subfam. nov. from Discussions to Results.
Remove underlining for some Kirejtshuk in References.
I think that English is very good, but needs minor editing.
Author Response
Many thanks for the positive review. I changed our manuscript following the mentioned recommendations and improve the English in it. The evidence of sexual attribution is explained by presence of the anal sclerite in the specimen examined. An additional Division with results of the poaper is not necessary because the results are clearly shown in the general composition of the paper.
Round 2
Reviewer 1 Report
From what I gather from their answer, the authors counter a phylogeny based on the classical evolutionistic systematics (i.e. Mayr and Simpson) based on “real fact and their logic interpretation”, that they follow, to the cladistic phylogeny or phylogenomics “which joint facts and ideas without factual base”. I have not understood if their strong criticism concerns only the paper by Cai et al. [10] or cladistics and molecular studies in general. However, this is not the place for a discussion about this complex subject. As I previously wrote, this paper deserves publication just because it will surely elicit interesting debate, at least I hope.